# Prevalence of HIV infection and related risk factors among young Thai men between 2010 and 2011

Julius Eleazar dC. Jose[1,2], Boonsub Sakboonyarat[3], Khunakorn Kana[4], Thippawan Chuenchitra[4], Akachai Sunantarod[5], Supanee Meesiri[4], Mathirut Mungthin[6], Kenrad E. Nelson[7], Ram Rangsin[3]*

1 Graduate Program in Biomedical Science, Faculty of Allied Health Sciences, Thammasat University, Klong Luang, Pathum Thani, Thailand, 2 Department of Medical Technology, Faculty of Pharmacy, University of Santo Tomas, Sampaloc, Manila, Philippines, 3 Department of Military and Community Medicine, Phramongkutklao College of Medicine, Bangkok, Thailand, 4 Armed Forces Institute of Medical Sciences (AFRIMS), Bangkok, Thailand, 5 Royal Thai Army Institute of Pathology (AIP), Bangkok, Thailand, 6 Department of Pharmacology, Phramongkutklao College of Medicine, Bangkok, Thailand, 7 Department of Epidemiology, Johns Hopkins University Bloomberg School of Public Health, Baltimore, Maryland, United States of America

* r_rrangsin@yahoo.com

**Data Availability Statement:** The study has been reviewed and approved by the Institutional Review Board, Royal Thai Army Medical Department in compliance with international guidelines such as

## Abstract

### Introduction

Understanding the current epidemiology of human immunodeficiency virus (HIV) infection in Thailand will facilitate more effective national HIV prevention programs. This study aimed to determine the prevalence and risk factors for HIV infection among young Thai men.

### Methods

A total survey was conducted of Royal Thai Army new conscripts, participating in the national HIV surveillance in November 2010 and May 2011. Behavioral risk factors for HIV infection were determined using a standardized survey questionnaire in the total study population and men who have sex with men (MSM) subgroup.

### Results

A total of 301 (0.5%) HIV infected young Thai men were identified from the total study population (63,667). Independent risk factors associated with HIV infection among the total study population included being single (adjusted Odds Ratio [AOR] 1.6, 95% Confidence Interval [CI] 1.1–2.2), having no formal education (AOR 6.5, 95% CI 2.3–18.4) or a bachelor's degree (AOR 1.8, 95% CI 1.0–3.0), engaging in bisexual (AOR 3.7, 95% CI 2.4–5.6) or exclusively homosexual activity (AOR 14.4, 95% CI 10.4–19.8), having a history of Sexually Transmitted Infection (STI) (AOR 2.3, 95% CI 1.6–3.3) and having sex in exchange for gifts/money (AOR 2.0, 95% CI 1.5–2.8). A total of 4,594 (7.9%) MSM were identified, of which 121 (2.6%) were HIV infected. The prevalence of HIV infection among MSM in urban (2.8%) and rural (2.4%) areas were relatively comparable (p-value = 0.44). Of the identified MSM,

Declaration of Helsinki, the Belmont Report, CIOMS Guidelines and the International Conference on Harmonization of Technical Requirements for Registration of Pharmaceuticals for Human Use – Good Clinical Practice (ICH - GCP). Data cannot be shared publicly because the data set contains sensitive identifying information including HIV status; thus, there are ethical restrictions on the data set. Data are available from the Research Unit for Military Medicine, Phramongkutklao College of Medicine, Bangkok, Thailand (contact via pcmmc@pcm.ac.th) for researchers who meet the criteria for access to confidential data. The data set names, and some variables are provided in the Supporting Information files.

**Funding:** Thailand MOPH – U.S. CDC Collaboration (TUC), Thailand Ministry of Public Health Bureau of Epidemiology and Bureau of AIDS, Tuberculosis, and Sexually Transmitted Infections Award Number:
None | Recipient: Ram Rangsin, M.D., M.P.H., Dr. P.H. The Scholarships for Foreign Students Studying for a Degree 2559 B.E., Thammasat University. Award Number: None | Recipient: Julius Eleazar dC. Jose.

**Competing interests:** The authors have declared that no competing interests exist.

82.5% reported having sexual desire with females only. Risk factors associated with HIV infection in the MSM subgroup included living in the western region (AOR 3.5, 95% CI 1.2–10.4), having a bachelor's degree (AOR 2.7, 95% CI 1.2–5.7), having a history of exclusive receptive (AOR 3.6, 95% CI 1.6–7.7) or versatile anal sex (AOR 4.7, 95% CI 3.0–7.5) and history of having sex in exchange for gifts/money (AOR 2.3, 95% CI 1.5–3.5).

## Conclusion

The prevalence of HIV infection among young Thai men has continued to be below 0.5% in 2010 and 2011. High risk sexual activity, including MSM, played a major role in the HIV epidemic among this population. Effective HIV prevention programs should cover MSM who have heterosexual desire as well as having sex in exchange for gifts/money and be implemented in both urban and rural areas.

## Introduction

The marked decline in the human immunodeficiency virus (HIV) epidemic in Thailand has been attributed to identifying and controlling its associated behavioral risk factors. [1, 2] It was estimated that 440,000 people are living with HIV or Acquired Immunodeficiency Syndrome (AIDS) in Thailand at the end of 2017 as compared with 570,000 at the end of 2005. [3] Though the HIV epidemic started with people who inject drugs, a dramatic increase was observed when the infection reached female sex workers (FSWs) and their clients. During the peak of the epidemic, the spread of infection was strongly associated with heterosexual transmission, more specifically among sexually active males having unprotected sex with an FSW. This practice led to the lateral transmission to their female sexual partners and further to their children. [4] Identifying these risk factors led to the move to promulgate protected sex by implementing a national 100% condom use program [5] that subsequently changed the sexual behavior of young Thai men. Between 1991 and 1995, the proportion of young Thai military conscripts in northern Thailand who had sex with a FSW dropped from 81.4 to 63.8%; condom use increased from 61.0 to 92.6%; while the prevalence of HIV infection decreased to 6.7% from a high of 12.5%. [6–9] The implementation of the program also considerably changed the prevalence of HIV infection in the targeted population, venue-based FSWs. In Bangkok, from 1993 to 1996, a 48% decline has been found in sex with FSWs and an increase in the use of condoms among male attenders of sexually transmitted disease (STD) clinics. [10] In terms of treatment, the National Access to Antiretrovirals Program for People Living with HIV/AIDS (NAPHA) and the use of antiretroviral therapy (ART) combinations since the early 2000s has decreased viral transmission and increased the quality of life among HIV-infected patients by providing free of charge ART. [11–16]

In the early 2000s, the heterosexual transmission had been contained resulting in a significant decline in the national incidence of HIV infection. [12] However, HIV transmission among MSM is growing rapidly. In 2005, this transmission route contributed 22.6% of new infections and would be 33% in 2010 and 43% by 2015. [17] A significant increase in the estimated HIV incidence among MSM was observed from 4.1% in 2003 to 7.7% in 2007 [12, 18] to 9.2% in 2014 [12]. In 2015, six community-based clinics in the provinces of Bangkok, Chiang Mai, Chonburi and Songkhla, catering to MSM and transgender women, reported the prevalence of HIV infection among MSM at 15% and transgender women at 8.8%. HIV

infection, among MSM in the study, was associated with behavioral risk factors including having a history of sexually transmitted infection (STI) and moderate to high risk of acquiring HIV infection at baseline. [19]

Young Thai men, conscripted in the Royal Thai Army (RTA) aged 17 to 29 years, represent a national population of young Thai men as a sample. [9, 20] We determined the prevalence of HIV infection and related risk factors among newly inducted young Thai men conscripted in the RTA nationwide in the November 2010 and May 2011 rounds. Data from the total survey could provide essential information to monitor HIV infection including associated risk factors that would be vital to fine-tuning and developing national HIV prevention programs in the country.

## Methods

### Study designs and subjects

The RTA holds an annual selection of young men aged 21 years for conscription in April at the district level of their home province. Exemptions are given to a subset of men who are either disabled or severely ill, transgender women (TG) and individuals who participate in alternative military service including the Thai Reserve Officer Training Corps Student (TROTCS) program. In Thailand, the government allows approximately 100,000 secondary school and university students to participate in the TROTCS program as an alternative military service each year. Those students who complete the TROTCS program would be excluded from the conscription process when they reach 21 years old. This exemption does not exclude individuals with asymptomatic HIV infection nor exclude individuals based on their sexual orientation or drug use. Individuals who do not participate in the selection process without a valid exception face legal sanctions and penalties. Therefore, this makes the lottery system completely autonomous, random and uniform in producing a reliable mechanism for sampling young men throughout the whole country. Those chosen will enter military service in either May or November of the same year [21]. Since 2001, young men aged 17 to 20 or 22 years or older are accepted as volunteers without passing through the lottery system. [9, 22]

HIV surveillance among military conscripts, part of the national HIV surveillance program, started in 1989. This includes serological testing for HIV and a short demographics questionnaire that does not include any behavioral risk factor questions. The collection of blood samples and the deployment of the short demographic questionnaire were supervised by competent personnel from local military hospitals of each base. The blood samples were processed at the Army Institute of Pathology (AIP) while the questionnaires were processed at the Armed Forces Institute of Medical Sciences (AFRIMS), both in Bangkok. This surveillance activity was scheduled during the first two weeks after induction. Blood samples were collected after HIV pretest counseling and after obtaining informed consent to participate in the surveillance activity. Blood samples were then tested for HIV antibody using the enzyme-linked immunosorbent assay (ELISA) and confirmed using the Western Blot test. Confirmation was made using a second sample to ensure the reliability of results. Tests results were then released to the identified HIV-infected conscripts through designated physicians or trained nurses, who in turn would provide posttest counseling approximately 4 to 5 months after induction. Confidentiality of the results was ensured by limiting the number of individuals that would handle the transfer of information from the laboratory to the HIV-infected conscript. HIV-infected conscripts remained in military service unless their health status became an impediment. Of the total number of conscripts entering RTA military service in November 2010 and May 2011, 67,170 (97.6%) participated in the national HIV surveillance activity. These men served as the target population of the study, regardless of HIV serologic status. Inclusion

criteria consisted of men who (a) were 18 years of age or older and (b) gave informed consent. Exclusion criterion was the new RTA conscripts who were unable to answer the self-administrated questionnaire during the specified dates in each military camp.

## Data collection

Because the National HIV surveillance program uses only a short demographic questionnaire; we, therefore, created a more detailed questionnaire including questions on behavioral risk factors for HIV infection. After the written informed consent process, the enrolled study participants were asked to complete self-administered questionnaires in the private environment in their camps during the first eight weeks of the basic military training. These questionnaires were completed before the HIV test results were reported to the men to avoid information bias. All completed questionnaires were sent directly from each training unit to the data management unit in Bangkok. Anonymity was ensured using unique codes that could only be decoded by the respective data management personnel. The questionnaire was used to identify risk factors to HIV infection that were of interest and had been developed from related risk factors studies among young Thai men. Men having sex with men status was defined as the lifetime sexual activity of a man having sex with another man and not as the identity expressed by the person. Lifetime sex partner consisted of the number of sex partners including the types of sexual partners. History of intravenous drug use, history of incarceration, history of HIV testing, history of sex with a FSW, history of sex in exchange of gifts/money, history of sexual coercion, and sexual preference were all defined as the occurrence of the specific events in their lifetime. Sexual experience including the classification of MSM status was inferred from the responses to different but related questions in the questionnaire. Sexual transmitted infection was defined as the participants having a history of STIs in the previous 12 months before induction. Condom use with a female sex worker was defined as the participants who have a history of condom use with a FSW in the last 12 months.

## Statistical analysis

The responses of the participants to the self-administered risk factor survey were encoded in a computer-based program with their HIV serostatus added at the end of the process. The data were analyzed in two stages: first as a whole representing a cohort of young Thai men conscripted in the RTA, and second, MSM as a subgroup identified based on their responses in the risk factor survey representing a cohort of MSM.

Appropriate measures to determine central tendencies were used to describe continuous data, and percentage was used for categorical data. To compare the effects of potential risk factors for HIV infection across HIV serostatus, the $\chi 2$ or Fisher's exact test was used for categorical variables while the student's t-test was used for continuous variables. The odds ratio and 95% confidence intervals (CI) of both demographic and behavioral variables associated with HIV infection were analyzed using univariate analysis. A multiple logistic regression model was used to determine the independent effects of significant risk factors. Statistical significance was determined using the 0.05 cut off for the $p$-value.

## Ethics consideration

The study protocol was approved by the Institutional Review Boards of the RTA Medical Department and the Ethics Subcommittee of Thammasat University. Written informed consent forms were acquired before enrolling the participants. Identified HIV-infected participants were given post-test counseling and treatment following standard guidelines in Thailand.

## Results

### Demographic characteristics

A total of 67,170 young Thai men conscripted in the RTA in November 2010 and May 2011 participated in the national HIV sero-surveillance comprising the baseline population for this national risk factor survey. Of this total number of conscripts, 63,667 (94.8%) participated in this study: 27,672 (43.5%) were from the November 2010 round and 35,995 (56.5%) were from the May 2011 round of induction. The participants were invited from 330 RTA basic military units nationwide. Demographics and behavioral risk factor profiles of the participants before induction are summarized in Table 1.

The average age of the participants was 21.4 years (±1.0). Regarding demographic profile before induction; 3.6% of these men did not live with family, 36.2% lived in the northeast region two years before induction, 70.9% were single and 5.8% had obtained a bachelor's degree. Concerning reported behavioral risk factors for HIV infection, the average age of sexual debut was 16.6 (±2.0) years, 93.2% had engaged in some form of a sexual act and with an average of 5.6 (±6.3) lifetime sexual partners. Moreover, 34.6% had a history of sex with an FSW, 7.9% had a history of sex with another man, 5.8% had a history of STI and 7.3% had a history of providing sex in exchange for gifts/money. In terms of sexual activity, 92.2% were exclusively heterosexuals, 4.8% were bisexuals and 3.1% were exclusively homosexuals. History for HIV testing in lifetime and 12 months accounted for 26.8% and 8.2%, respectively.

### Prevalence of HIV infection

A total of 301 (0.5%) young Thai men were identified to be HIV-infected. Of 4,589 identified MSM, 121 (2.6%) were identified to be HIV-infected cases. This corresponded to 40.2% of the total number of HIV cases in the study.

### Risk factors of HIV infection among young Thai men

Risk factors for HIV infection are summarized in Table 2. The risk factors that were independently associated with HIV infection among the young Thai men includes having exclusively homosexual activity (AOR: 14.4; 95%CI: 10.4–19.8) or bisexual (AOR: 3.7; 95%CI: 2.4–5. 6), having no formal education (AOR: 6.5; 95%CI: 2.3–18.4) or a bachelor's degree (AOR: 1.8; 95%CI: 1.0–3.0), having a history of STI (AOR: 2.3; 95%CI: 1.6–3.3), providing sex in exchange for gifts/money (AOR: 2.0; 95%CI: 1.5–2.8), and being single (AOR: 1.6; 95% CI: 1.1–2.2) after adjusting for age, living status and region of residence two years before induction. Having a history of sex with an FSW (AOR: 0.6; 95% CI: 0.4–0.8) was inversely associated with HIV infection in the overall study population.

We had the opportunity to identify MSM based on their reported homosexual activity. From the total number of participants, 4,589 (7.9%) were identified to have had a history of sex with another man. Because MSM were identified from the total of young Thai men, we were able to differentiate MSM and non-MSM based on their demographic and behavioral profiles, as summarized in Table 3.

The average age of MSM was 21.4 (±1.0) years, comparable to that of non-MSM. In terms of place of residence two years before induction, we found that almost 40% of MSM lived in a rural area in the province of their residence. Having a history of anal sex was reported at 91.8% of MSM of which 69.9% engaged in exclusively insertive anal sex, 4.4% involved exclusively receptive anal sex, and 25. 8% involved versatile sex. According to sexual activity, 60.5% practiced bisexual while 39.5% practiced exclusively homosexual activities. In terms of sexual preference or sexual desire, 82.4% preferred to have sex with females only while 11.1% preferred to

**Table 1. Demographic and behavioral profile of the participants before induction.**

| Characteristics | N = 63,667 (%) |
|---|---:|
| **Age, yrs** | |
| Mean (SD) | 21.35 (±1.00) |
| **Round of induction** | |
| November 2010 | 27,672 (43.5) |
| May 2011 | 35,995 (56.5) |
| **Living with family**[a] | |
| No | 2,228 (3,6) |
| Yes | 60,293 (96.4) |
| **Region of residence 2 years before induction** | |
| Upper North | 6,543 (10.6) |
| Lower North | 3,995 (6.5) |
| Northeast | 22,363 (36.2) |
| East | 3,834 (6.2) |
| Central and West | 11,113 (18.0) |
| Bangkok | 5,302 (8.6) |
| South | 8,641 (14.0) |
| **Area of residence** | |
| Urban | 36,415 (58.5) |
| Rural | 25,875 (41.5) |
| **Occupation** | |
| Employee | 36,971 (58.4) |
| Student | 10,186 (16.1) |
| Agricultural | 10,637 (16.8) |
| Unemployed | 5,522 (8.7) |
| **Marital status** | |
| Single | 43,717 (70.9) |
| Married | 17,907 (29.1) |
| **Educational attainment** | |
| No formal | 226 (0.4) |
| Grade 1 to Grade 9 | 36,110 (57.0) |
| Grade 10–12 and Vocational | 23,289 (36.8) |
| Bachelor's degree | 3,690 (5.8) |
| **History of injecting drug use** | 2,241 (3.6) |
| **History of non-injecting drug use** | 31,813 (50.0) |
| **History of incarceration**[*] | 3,484 (10.2) |
| **History of previous HIV testing (lifetime)** | 15,393 (26.8) |
| **History of blood transfusion** | 3,520 (5.6) |
| **Circumcised** | 6,768 (11.0) |
| **Average age at first sex (years)** | 16.6 (±2.0) |
| **History of sex with a female sex worker** | 20,543 (34.6) |
| **Number of lifetime sex partner** | 5.6 (±6.3) |
| **History of sex with another man** | 4,589 (7.9) |
| **Sexual experience (Lifetime)** | |
| Exclusively heterosexual | 53,845 (92.2) |
| Bisexual | 2,781 (4.8) |
| Exclusively homosexual | 1,808 (3.1) |
| **Sexual preference/desire** | |

(*Continued*)

**Table 1.** (Continued)

| Characteristics | N = 63,667 (%) |
|---|---|
| Female only | 61,824 (98.3) |
| Both Male and Female | 661 (1.1) |
| Male only | 391 (0.6) |
| **History of sexually transmitted infection** | 3,337 (5.8) |
| **History of sex in exchange for gifts/money** | 4,313 (7.3) |
| **History of sexual coercion** | 3,160 (5.1) |
| **Condom use with a female sex worker in the last 12 months** | |
| Always | 8,456 (77.4) |
| **HIV infected cases** | 301 (0.47) |

*Data from May 2011 round of induction only.

<sup>a</sup>Living with family; parents, wife/lover and relatives.

have sex with both males and females and 6.4% preferred to have sex exclusively with males. Of the 4,589 MSM, 31.8% reported a same sex experience in the previous 12 months. The median number of male sex partners was two individuals, while the median number of male sex partners in their lifetime was three individuals.

### Risk factors of HIV infection among MSM

Risk factor analysis for HIV infection among MSM is summarized in Table 4. The identified risk factors for HIV infection among MSM includes history of versatile (AOR: 4.7; 95% CI: 2.9–7.5) or exclusively receptive (AOR: 3.6; 95% CI: 1.6–7.7) anal sex (as compared with exclusively insertive), living in the western region two years before induction (AOR: 3.5; 95% CI: 1.2–10.4), higher educational attainment (AOR: 2.7; 95% CI: 1.2–5.7), and history of sex in exchange for gifts/money (AOR: 2.3; 95% CI: 1.5–3.5). Having a history of sex with an FSW (AOR: 0.3; 95% CI: 0.2–0.5) was found to be inversely associated with HIV infection.

Having a history of sex with an FSW was initially found to be inversely associated with HIV infection in both the total population of young Thai men and the MSM subgroup. However, when we analyzed the data to identify the association between history of sex with an FSW and HIV infection stratified by MSM status, we found that a history of sex with an FSW was no longer a risk factor for HIV infection among non-MSM (AOR: 1.1; 95% CI: 0.8–1.6).

### Discussion

Our data demonstrated patterns of sexual behaviors and risk factors for HIV infection among young Thai men including MSM from a total survey of newly inducted RTA conscripts. We reported that the prevalence of HIV infection among young Thai men from 2010 to 2011 was 0.5%. This extends the trends of prevalence of HIV infection below 1% since the 2000s. [9, 23, 24] Furthermore, the prevalence of HIV infection among young Thai men residing in Bangkok, the eastern region, and upper and lower northern regions were higher than 0.5%.

From our current study, we found that sex between men likely played a major role in the recent HIV epidemic among young Thai men. Of the 301 HIV-infected study participants, 121 (40.2%) men reported a history of having sex with another man. The overall prevalence of HIV infection in the young MSM subgroup of our study was 2.63%. Moreover, MSM from the following regions of the country had prevalences of HIV infection higher than the overall prevalence including the upper north (4.8%), the west (3.6%) and Bangkok (4.7%). When we

**Table 2. Univariate and multivariate analysis of risk factors to HIV infection among the participants inducted into the RTA in November 2010 and May 2011.**

| Characteristics | Total | HIV+ (%) | Crude OR (95% CI) | Adjusted OR (95% CI) |
|---|---|---|---|---|
| **Age (years)** | | | | |
| Mean (SD) | 21.4 (±1.0) | 21.5 (±1.0) | 1.1 (1.0–1.2) | 1.0 (0.8–1.1) |
| **Living with family**[a] | | | | |
| No | 2,228 | 28 (1.3) | 2.8 (1.9–4.2) | 1.5 (1.0–2.5) |
| Yes | 60,293 | 267 (0.4) | 1.0 | 1.0 |
| **Region of residence 2 years before induction** | | | | |
| Upper North | 6,543 | 37 (0.6) | 1.9 (1.1–3.1) | 1.4 (0.8–2.3) |
| Lower North | 3,995 | 24 (0.6) | 2.0 (1.2–3.5) | 1.1 (0.6–2.1) |
| Northeast | 22,363 | 88 (0.4) | 1.3 (0.9–2.0) | 1.0 (0.6–1.6) |
| East | 3,834 | 24 (0.6) | 2.1 (1.2–3.6) | 1.5 (0.8–2.7) |
| Central and West | 11,113 | 52 (0.5) | 1.6 (1.0–2.5) | 1.0 (0.6–1.7) |
| Bangkok | 5,302 | 39 (0.7) | 2.5 (1.5–4.0) | 1.3 (0.8–2.3) |
| South | 8,641 | 26 (0.3) | 1.00 | 1.00 |
| **Area of residence** | | | | |
| Urban | 36,415 | 179 (0.5) | 1.1 (0.9–1.4) | - |
| Rural | 25,875 | 116 (0.5) | 1.00 | - |
| **Occupation** | | | | |
| Student | 10,186 | 60 (0.6) | 2.0 (1.3–3.1) | - |
| Employee | 36,971 | 189 (0.5) | 1.8 (1.2–2.6) | - |
| Unemployed | 5,522 | 20 (0.4) | 1.2 (0.7–2.2) | - |
| Agriculture | 10,637 | 31 (0.3) | 1.00 | - |
| **Marital status** | | | | |
| Single | 43,717 | 233 (0.5) | 1.7 (1.3–2.2) | 1.6 (1.1–2.2) |
| Married | 17,907 | 57 (0.3) | 1.0 | 1.0 |
| **Educational attainment** | | | | |
| No formal | 226 | 5 (2.2) | 4.7 (1.9–11.6) | 6.5 (2.3–18.4) |
| Grade 1 to Grade 9 | 36,110 | 172 (0.5) | 1.0 | 1.0 |
| Grade 10–12 *incl.* Vocational | 23,289 | 90 (0.4) | 0.8 (0.6–1.1) | 1.0 (0.7–1.3) |
| Bachelor's degree | 3,690 | 32 (0.9) | 1.8 (1.3–2.7) | 1.8 (1.0–3.0) |
| **History of injecting drug use** | | | | |
| Yes | 2,241 | 17 (0.8) | 1.7 (1.0–2.8) | - |
| No | 60,459 | 277 (0.5) | 1.0 | - |
| **History of sex with a female sex worker** | | | | |
| Yes | 20,543 | 93 (0.5) | 0.9 (0.7–1.2) | 0.6 (0.4–0.8) |
| No | 38,887 | 196 (0.5) | 1.0 | 1.0 |
| **Sexual experience** | | | | |
| Exclusive heterosexual | 53,845 | 167 (0.3) | 1.0 | 1.0 |
| Bisexual | 2,781 | 33 (1.2) | 3.9 (2.7–5.6) | 3.7 (2.4–5.6) |
| Exclusive homosexual | 1,808 | 88 (4.9) | 16.4 (12.7–21.4) | 14.4 (10.4–19.8) |
| **Sexual preference/desire** | | | | |
| Female only | 61,824 | 179 (0.3) | 1.0 | - |
| Male only | 391 | 56 (14.3) | 52.3 (38.1–71.7) | - |
| Both male and female | 661 | 44 (6.7) | 22.3 (15.9–31.2) | - |
| **History of sexually transmitted infection** | | | | |
| Yes | 3,337 | 47 (1.4) | 3.4 (2.4–4.6) | 2.3 (1.6–3.3) |
| No | 54,304 | 230 (0.4) | 1.0 | 1.0 |
| **History of sex in exchange for gifts/money** | | | | |

*(Continued)*

**Table 2.** (Continued)

| Characteristics | Total | HIV+ (%) | Crude OR (95% CI) | Adjusted OR (95% CI) |
|---|---|---|---|---|
| Yes | 4,313 | 78 (1.8) | 4.8 (3.7–6.2) | 2.0 (1.5–2.8) |
| No | 54,619 | 211 (0.4) | 1.0 | 1.0 |
| **History of sexual coercion** | | | | |
| Yes | 3,160 | 42 (1.3) | 3.1 (2.2–4.3) | - |
| No | 58,404 | 253 (0.4) | 1.0 | - |

[a]Living with family; parents, wife/lover and relatives.

compared the prevalence of HIV infection among MSM from the current study with the other reports, we found that several studies reported a relatively higher prevalence of HIV infection. A study among MSM and transgender women in six community-based clinic sites in Bangkok, Chiang Mai, Chonburi and Songkhla reported a prevalence of HIV infection of 15.0% among MSM at baseline in 2016. [19] Two studies on HIV infection in a community clinic in Bangkok reported the prevalence of HIV infection among Thai MSM at 21% in 2010. [25, 26] These studies used venue-based enrollment in a high risk MSM population aged ≥18 years. However, our study focused on a homogenous population of young Thai men aged 21 years, reported having a history of sex with another man, nationwide.

One of the major findings related to MSM sexual activity from our study was that the majority (82.4%) of those having sex with another man reported having heterosexual desire. Those MSM having heterosexual desire had a substantially lower risk for HIV infection compared with those MSM who were exclusively homosexual and bisexual. In addition, approximately one-third of the MSM study population reported having a history of sex in exchange for gifts/money. Related studies have noted that the majority of MSM who engage in sex in exchange for gifts/money were predominantly heterosexual or bisexual. [27, 28]

Reported history of sex with an FSW was identified as one of the prime movers in the HIV epidemic in Thailand in the early 90s. [4] However, after implementing the 100% condom use program in Thailand, targeting FSWs and their clients, a decline in the prevalence of HIV infection corresponding to an increase in condom use during sexual intercourse with an FSW was observed. [6, 29–32]

Our study report that having a history of sex with a FSW was no longer associated with HIV infection in the subgroup of non-MSM conscripts in 2010 and 2011. This reflects the continued decline in the prevalence of HIV infection among FSW and their clients suggesting the continued effectiveness of the 100% condom use program among institution based FSW. Notably, while the proportion of young Thai men who reported having sex with an FSW had declined to less than 40% in 2009, the proportion of men who reported having sex with a girlfriend increased [6, 9, 23]

We found that having a history of sex with an FSW was inversely associated with HIV infection among overall study participants. When we analyzed the data to identify the association between a history of sex with a FSW and HIV infection stratified by MSM status, we found that in a non-MSM study population having a history of sex with a FSW was no longer associated with HIV infection in 2010 and 2011. However, a history of sex with an FSW was still inversely associated with HIV infection in the MSM subgroup. One of the explanations for this finding is that a history of sex with a FSW may be a proxy indicator for having a heterosexual preference in our study population. This explanation is also supported by our findings that the prevalence of HIV infection among those MSM reporting that they had sexual desire with

**Table 3. Demographic and behavioral profile of identified men who have sex with men and non-MSM among the participants before induction.**

| Characteristics | MSM (%) N = 4,589 | Non-MSM (%) N = 53,845 | p-value |
|---|---|---|---|
| **Round of induction** | | | 0.021[c] |
| November 2010 | 2,103 (45.8) | 23,732 (44.1) | |
| May 2011 | 2,486 (54.2) | 30,113 (55.9) | |
| **Age, years** | | | 0.562 [a] |
| Mean (SD) | 21.4 (±1.0) | 21.4 (±1.0) | |
| **Living with family[d]** | | | <0.001[c] |
| No | 249 (5.6) | 1,805 (3.4) | |
| Yes | 4,236 (94.4) | 51,153 (96.6) | |
| **Region of residence 2 years before induction** | | | <0.001[a] |
| Upper North | 334 (7.5) | 5,610 (10.7) | |
| Lower North | 332 (7.5) | 3,408 (6.5) | |
| North East | 1,574 (35.5) | 18,754 (35.8) | |
| East | 257 (5.8) | 3,295 (6.3) | |
| Central | 776 (17.5) | 7,699 (14.7) | |
| West | 196 (4.4) | 1,663 (3.2) | |
| South | 470 (10.6) | 7,478 (14.3) | |
| Bangkok | 492 (11.2) | 4,476 (8.6) | |
| **Area of residence** | | | 0.009 [c] |
| Urban | 2,707 (60.5) | 30,829 (58.5) | |
| Rural | 1,771 (39.5) | 21,893 (41.5) | |
| **Occupation** | | | <0.001[b] |
| Employee/Factory worker | 1,329 (32.6) | 16,070 (33.1) | |
| Agriculture/Fisherman | 640 (15.7) | 8,812 (18.2) | |
| Student | 524 (12.9) | 8,669 (17.9) | |
| Unemployed | 527 (12.9) | 4,537 (9.4) | |
| Laborer | 467 (11.4) | 4,072 (8.4) | |
| Own business | 371 (9.1) | 4,242 (8.7) | |
| Sales | 222 (5.5) | 2,097 (4.3) | |
| **Marital status** | | | <0.001[c] |
| Single | 3,237 (72.8) | 35,876 (68.7) | |
| Married | 1,209 (27.2) | 16,332 (31.3) | |
| **Educational attainment** | | | <0.001[b] |
| No formal | 25 (0.6) | 158 (0.3) | |
| Grade 1 to Grade 9 | 2,991 (65.5) | 30,350 (56.7) | |
| Grade 10–12 and Vocational | 1,343 (29.3) | 19,956 (37.2) | |
| Bachelor's Degree | 210 (4.6) | 3,118 (5.8) | |
| **History of injecting drug use** | | | 0.001[c] |
| Yes | 200 (4.4) | 1,826 (3.4) | |
| No | 4,317 (95.6) | 51,296 (96.6) | |
| **History of non-injecting drug use** | | | <0.001[c] |
| Yes | 2,980 (64.9) | 27,322 (50.7) | |
| No | 1,609 (35.1) | 26,523 (49.3) | |
| **History of incarceration*** | | | <0.001[c] |
| Yes | 447 (19.0) | 2,841 (9.9) | |
| No | 1,902 (81.0) | 51,842 (90.1) | |
| **Age (years) at first sexual intercourse** | | | <0.001[a] |
| Mean (SD) | 16.0 (±2.0) | 16.6 (±2.0) | |

(*Continued*)

**Table 3.** (Continued)

| Characteristics | MSM (%) N = 4,589 | Non-MSM (%) N = 53,845 | p-value |
|---|---|---|---|
| **History of sex with a female sex worker** | | | <0.001[c] |
| Yes | 2,558 (57.0) | 17,494 (32.9) | |
| No | 1,928 (43.0) | 35,658 (67.1) | |
| **Number of lifetime sex partner** | | | <0.001[a] |
| Mean (SD) | 8.7 (±9.0) | 5.9 (±6.0) | |
| **Sexual experience (Lifetime)** | | | <0.001[b] |
| Exclusive heterosexual | - | 53,845 (100.0) | |
| Bisexual | 2,781 (60.6) | - | |
| Exclusive homosexual | 1,808 (39.4) | - | |
| **Sexual preference/desire** | | | <0.001[b] |
| Female only | 3,735 (82.4) | 53,209 (99.7) | |
| Male only | 288 (6.4) | 58 (0.1) | |
| Both male and female | 504 (11.1) | 100 (0.2) | |
| **History of sexually transmitted infection** | | | <0.001[c] |
| Yes | 633 (14.5) | 2,650 (5.1) | |
| No | 3,735 (85.5) | 48,736 (94.9) | |
| **History of sex in exchange for gift/money** | | | <0.001[c] |
| Yes | 1,421 (31.7) | 2,802 (5.3) | |
| No | 3,065 (68.3) | 49,736 (94.7) | |
| **History of sexual coercion** | | | <0.001[c] |
| Yes | 710 (15.7) | 2,342 (4.4) | |
| No | 3,812 (84.3) | 50,907 (95.6) | |
| **HIV infected cases** | 121 (2.6) | 167 (0.3) | <0.001[c] |

[*]Data available for May 2011 round of induction only.

[a]p-value for comparison of the mean of characteristic between groups (independent sample t-test).

[b]p-value for comparison of the proportion of characteristics between groups (Chi-square test).

[c]p-value for comparison of the proportion of characteristics between groups (Fisher's Exact test).

[d]Living with family; parents, wife/lover and relatives.

females only was 0.7%, while it was 19.1% among those MSM who reported that they had sexual desire with males only.

We identified that educational attainment was independently associated with HIV infection both in the total study population and the MSM subgroup. Those men who had obtained a bachelor's degree or higher were more likely to acquire HIV infection than those who were at grade 9 and lower educational level in both the total study population of young Thai men and the MSM subgroup. Our finding was consistent with the related study from 2005 to 2009 reporting that young Thai men with a college degree had a higher risk for HIV infection. [9] The higher risk for HIV infection among college graduated might be because the participants having college degrees had a higher proportion of sexual preference for males (4.1% in total participants, 58.6% in MSM subgroup) compared with those with lower than college education (1.5% in total participants, 15.6% in MSM subgroup). In our study, sexual preference was a relatively strong risk factor for HIV infection (with male only crude OR 52.5; 95%CI 38.1–71.7, with both male and female crude OR 22.3; 95%CI 15.9–31.2). This effect was also shown in the MSM subgroup population (with male only crude OR 32.4; 95%CI 20.1–52.4, with both male and female crude OR 11.5; 95%CI 7.0–19.0).

**Table 4. Univariate and multivariate analysis of risk factors to HIV infection among identified MSM inducted into the RTA in November 2010 and May 2011.**

| Characteristics | Total | HIV+ (%) | Crude OR (95% CI) | Adjusted OR (95% CI) |
|---|---|---|---|---|
| **Age (years)** | | | | |
| Mean (SD) | 21.4 (±1.0) | 21.4 (±1.0) | 1.2 (1.0–1.4) | 1.0 (0.8–1.2) |
| **Living with family[a]** | | | | |
| No | 249 | 15 (6.0) | 2.5 (1.4–4.4) | - |
| Yes | 4,236 | 105 (2.5) | 1.0 | - |
| **Region of residence 2 years before induction** | | | | |
| Central | 776 | 11 (1.4) | 1.0 | 1.0 |
| Upper North | 334 | 16 (4.8) | 3.5 (1.6–7.6) | 2.3 (0.9–5.6) |
| East | 257 | 7 (2.7) | 2.0 (0.8–5.1) | 1.6 (0.5–4.9) |
| West | 196 | 7 (3.6) | 2.6 (1.0–6.7) | 3.5 (1.2–10.4) |
| Bangkok | 492 | 23 (4.7) | 3.4 (1.7–7.0) | 2.0 (0.8–4.8) |
| Others | 2,376 | 56 (2.4) | 1.7 (0.9–3.2) | 1.5 (0.7–3.2) |
| **Area of residence** | | | | |
| Urban | 2,707 | 76 (2.8) | 1.2 (0.8–1.7) | 1.1 (0.7–1.7) |
| Rural | 1,771 | 43 (2.4) | 1.0 | 1.0 |
| **Occupation** | | | | |
| Laborer/Agriculture | 640 | 6 (0.9) | 1.0 | - |
| Student | 524 | 32 (6.1) | 6.9 (2.9–16.6) | - |
| Others | 3,402 | 83 (2.4) | 2.6 (1.2–6.1) | - |
| **Marital status** | | | | |
| Single | 3,237 | 106 (3.3) | 3.7 (2.0–6.9) | - |
| Married | 1,209 | 11 (0.9) | 1.0 | - |
| **Educational attainment** | | | | |
| No Formal to Grade 9 | 3,016 | 53 (1.8) | 1.0 | 1.0 |
| Grade 10–12 and Vocational | 1,340 | 44 (3.3) | 1.9 (1.3–2.9) | 1.7 (1.0–2.6) |
| Bachelor's degree | 210 | 23 (11.0) | 6.9 (4.1–11.5) | 2.7 (1.2–5.7) |
| **History of non-injecting drug use** | | | | |
| Yes | 2,980 | 52 (1.7) | 0.4 (0.3–0.6) | - |
| No | 1,609 | 69 (4.3) | 1.0 | - |
| **History of sex with a female sex worker** | | | | |
| Yes | 2,558 | 28 (1.1) | 0.2 (0.1–0.3) | 0.3 (0.2–0.5) |
| No | 1,928 | 93 (4.8) | 1.0 | 1.0 |
| **Sexual experience** | | | | |
| Bisexual | 2,781 | 33 (1.2) | 1.0 | - |
| Exclusive homosexual | 1,818 | 88 (4.8) | 4.3 (2.9–6.4) | - |
| **Sexual preference/desire** | | | | |
| Female only | 3,735 | 27 (0.7) | 1.0 | - |
| Male only | 288 | 55 (19.1) | 32.4 (20.1–52.4) | - |
| Both male and female | 504 | 39 (1.7) | 11.5 (7.0–19.0) | - |
| **History of anal sex with another man** | | | | |
| Yes | 3,727 | 113 (3.0) | 5.2 (1.3–21.1) | - |
| No | 334 | 2 (0.6) | 1.0 | - |
| **Type of anal sex with another man** | | | | |
| Insertive only | 2.371 | 31 (1.3) | 1.0 | 1.0 |
| Both Insertive and Receptive | 875 | 61 (7.0) | 5.7 (3.7–8.8) | 4.7 (2.9–7.5) |
| Receptive only | 148 | 10 (6.8) | 5.5 (2.6–11.4) | 3.6 (1.6–7.7) |
| **History of sexually transmitted infection** | | | | |

*(Continued)*

**Table 4.** (Continued)

| Characteristics | Total | HIV+ (%) | Crude OR (95% CI) | Adjusted OR (95% CI) |
|---|---|---|---|---|
| Yes | 633 | 24 (3.8) | 1.6 (1.0–2.5) | - |
| No | 3,735 | 92 (2.5) | 1.0 | - |
| **History of sex in exchange of gifts/money** | | | | |
| Yes | 1,421 | 55 (3.9) | 1.9 (1.3–2.7) | 2.3 (1.5–3.5) |
| No | 3,065 | 65 (2.1) | 1.0 | 1.0 |
| **History of sexual coercion** | | | | |
| Yes | 710 | 25 (3.5) | 1.4 (1.0–2.2) | - |
| No | 3,812 | 95 (2.5) | 1.0 | - |

[a]Living with family; parents, wife/lover and relatives.

Because our study population comprised newly inducted conscripts selected from every district nationwide; therefore, we had an opportunity to examine the effect of factors associated with HIV infection among men from all parts of the country. We found that study participants from an urban area had a comparable prevalence of HIV infection compared with those from a rural area in the total population of young Thai men and the MSM subgroup. However, several studies have shown that most preventive activities and research emphasized urban areas in the country. [6, 33–37] Our findings suggested that effective HIV prevention programs should be implemented not only in urban but also in the hard to reach populations in rural areas as well. Related reports from Canada found that people at risk of HIV infection, residing in the rural communities, had complex challenges [38] that significantly differed from urban areas including lack of access to health services and health information.

One of the limitations in our study related to the possibility of an underestimated prevalence of HIV infection among MSM. In Thailand, the military conscription process excluded young Thai men participating in an alternative military service such as the Thai Reserve Officer Training Corps Student (TROTCS), which involves approximately 20% of the total cohort of 21-years-old Thai men each year. Even though we did not have information on this MSM group in TROTCS, this group may have a higher prevalence of HIV infection because they had a higher education level. Those MSM with higher educational level may have a lower proportion of heterosexuals reporting to have had sex with another man compared with those MSM with lower educational level. The heterosexual men who experienced sex with another man had a lower risk for HIV infection. Some of the heterosexual men may have sex with another man in exchange for gift/money.

Because our study population was homogenous in terms of age distribution; therefore, we were unable to examine the effect of age on acquiring HIV infection in this study. The cross-sectional nature of the study may have limited explanations regarding temporal sequence associations from our study. Moreover, limitations related to the use of self-administered questionnaires that might not provide accurate answers especially concerning sensitive issues may have occurred.

## Conclusion

In conclusion, we reported epidemiological information regarding the prevalence and risk factors for HIV infection in a randomly selected national sample of young men conscripted in the RTA in November 2010 and May 2011. Our data suggested that sexual activity of MSM played a major role in the prevalence of HIV infection among young men in Thailand. We also found that having a history of sex with an FSW was no longer significantly associated with HIV

infection among young Thai non-MSM. Higher education level was found to be independently associated with HIV infection in the general population of young Thai men and the MSM subgroup. Finally, the risk of acquiring HIV infection among young Thai men residing in rural areas was comparable to the risk of acquiring HIV infection among those residing in urban areas. These data could be used to design prevention interventions for HIV infection that are more effective and evidence based.

## Supporting information

**S1 File.**
(PDF)

**S2 File.**
(PDF)

**S3 File.**
(PDF)

## Acknowledgments

We acknowledge the collaboration of the Royal Thai Army hospitals, Phramongkutklao College of Medicine Research and Development Office, Graduate Program of the Faculty of Allied Health Science–Thammasat University and all the participants.

## Author Contributions

**Conceptualization:** Julius Eleazar dC. Jose, Khunakorn Kana, Thippawan Chuenchitra, Akachai Sunantarod, Supanee Meesiri, Mathirut Mungthin, Kenrad E. Nelson, Ram Rangsin.

**Data curation:** Julius Eleazar dC. Jose, Thippawan Chuenchitra, Akachai Sunantarod, Supanee Meesiri.

**Formal analysis:** Julius Eleazar dC. Jose, Boonsub Sakboonyarat, Khunakorn Kana, Ram Rangsin.

**Methodology:** Julius Eleazar dC. Jose, Boonsub Sakboonyarat, Mathirut Mungthin, Ram Rangsin.

**Writing – original draft:** Julius Eleazar dC. Jose, Boonsub Sakboonyarat, Mathirut Mungthin, Kenrad E. Nelson, Ram Rangsin.

**Writing – review & editing:** Julius Eleazar dC. Jose, Boonsub Sakboonyarat, Mathirut Mungthin, Ram Rangsin.

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
