## [Decision Letter · Decision Letter 0]

11 May 2020

PONE-D-20-03591

Prevalence of HIV infection and related risk factors among young Thai men between 2010 and 2011

PLOS ONE

Dear Dr. Rangsin,

Thank you for submitting your manuscript to PLOS ONE. After careful consideration, we feel that it has merit but does not fully meet PLOS ONE’s publication criteria as it currently stands. Therefore, we invite you to submit a revised version of the manuscript that addresses the points raised during the review process.

Thank you for your submission. From my own reading of the manuscript, I agree with the reviewers comments and believe this manuscript will benefit from major revisions. Please review and carefully consider each of the comments below, incorporating those you feel appropriate.

We would appreciate receiving your revised manuscript by Jun 25 2020 11:59PM. To enhance the reproducibility of your results, we recommend that if applicable you deposit your laboratory protocols in protocols.io, where a protocol can be assigned its own identifier (DOI) such that it can be cited independently in the future. For instructions see: http://journals.plos.org/plosone/s/submission-guidelines#loc-laboratory-protocols

We look forward to receiving your revised manuscript.

Kind regards,

Ethan Morgan

Academic Editor

PLOS ONE

2. Please include additional information regarding the survey or questionnaire used in the study and ensure that you have provided sufficient details that others could replicate the analyses. For instance, if you developed a questionnaire as part of this study and it is not under a copyright more restrictive than CC-BY, please include a copy, in both the original language and English, as Supporting Information. In addition, please provide any details of the pre-testing of the questionnaire that took place.

"Funding: This work was supported by Thailand MOPH – U.S. CDC Collaboration (TUC), Thailand Ministry of Public Health Bureau of Epidemiology and Bureau of AIDS, Tuberculosis, and Sexually Transmitted Infections."

Reviewers' comments:

Reviewer's Responses to Questions

**Comments to the Author**

1. Is the manuscript technically sound, and do the data support the conclusions?

Reviewer #1: Yes

Reviewer #2: Partly

2. Has the statistical analysis been performed appropriately and rigorously? 

Reviewer #1: Yes

Reviewer #2: Yes

3. Have the authors made all data underlying the findings in their manuscript fully available?

Reviewer #1: Yes

Reviewer #2: Yes

4. Is the manuscript presented in an intelligible fashion and written in standard English?

Reviewer #1: Yes

Reviewer #2: Yes

5. Review Comments to the Author

Reviewer #1: In Methods :

1. please add inclusion and exclusion criteria of the subject

2. please add Operational definition of subject characteristics especially definition about the MSM and history of :

IVDU Incarceration, previous HIV Testing, Blood transfusion, sex with a female sex worker, number of lifetime sex partner, sex with another man, sexual experience, sexual preference, sexual transmitted infection, sex in exchange for gifts, sexual coercion, whether in 3, 6 or 12 months.

In Discussion:

1. Maybe you need more explain why having sex with a female sex worker have protective for the HIV risk (table. 2)

2. Why you concern about the non-inclusion of transgender women as the limitation of this study and it need to stated clearly in discussion.

Reviewer #2: 1. Does the manuscript address an important or timely issue?

Yes very much so, public health data on MSM and HIV care and treatment in Southeast Asia is very much needed.

2. Is the methodology used sound and the conclusions drawn valid?

Yes, I believe that the methodology and conclusions are valid.

Abstract

1. Second sentence, “The proportion of….” seems misplaced. I know you want to highlight this data, but it seems premature to place it second, whereas you repeat the same point below in the 2nd half of results.

2. What is the level of significance between urban and rural? Important for the abstract.

3. First sentence of Conclusion does not accurately capture the overall point of the study.

Background

1. Human immunodeficiency virus should NOT be capitalized

2. Line 53: in whom is that data reported on?

3. Line 65: very nonspecific, what does it mean to be “a shift to transmission”? This data is available and can be expressed as a proportion.

4. STI is not capitalized either

Methods

1. First sentence is passive, needs to be more active tense and specific

2. How can conscripts answer any of the questions described?

3. “Living alone or with friends” are quite different. What is the other option? Family?

Results:

1. Need data about the number of times each male had ever been HIV tested.

2. Data needs to be drilled down, in other words those men who have a same sex experience, how often was that? When was the last time they had MSM experience? How frequent was the contact?

3. Re-order the results to be from strongest association to weakest.

4. Discuss HIV infection prevalence in a separate paragraph.

5. Line 173 and following: where is the 7.86% data about a history of sex with another man?

6. I would not call sex with an FSW “protective” but rather has an inverse association.

7. Paragraph starting at line 200: too much detail, regurgitation of the tables.

Discussion

1. Valid discussion points

2. The association between education and risk through social media is tenuous at best, with no evidence. “Might be enhanced” is not evidence. Please see the findings from this study among Thai MSM:

Piyaraj P, van Griensven F, Holtz TH, Nelson K. The impact of internet use to recruit sex partners, and methamphetamine use on incident HIV infection among men who have sex with men, Bangkok, Thailand. Lancet HIV 2018;5(7):e379-e389. https://doi.org/10.1016/S2352-3018(18)30065-1. PMCID 6452023.

3. Were the participants randomly selected? Or were they part of a population-based sample of young men in Thailand? Any possible source of bias?

4. Also, they were not selected young men and MSM< rather just “men”?

- Minor Essential Revisions

1. In Discussion, “reported” or “report” in this manuscript? Since you are reporting now, would suggest not using the past tense. This holds true throughout the Discussion, i.e. line 271, etc

2. Discussion: sex between men LIKELY played a role….

3. Prefer “HIV-infected” instead of “HIV-positive”

4. Line 274: Diverted sexual partners, unclear meaning

5. Spelling of Acknowledgments

- Discretionary Revisions

These are suggested improvements, which the author may choose to consider.

1. Although you have the degrees of freedom to report 2 digits to the right of the decimal, this level of detail does not really aid the reader, 40.2% is as useful as 40.19%.

6. PLOS authors have the option to publish the peer review history of their article (what does this mean?). If published, this will include your full peer review and any attached files.

Reviewer #1: Yes: I Ketut Agus Somia

Reviewer #2: Yes: Timothy H. Holtz

---

## [Author Response · Author response to Decision Letter 0]

1 Jul 2020

PONE-D-20-03591

Prevalence of HIV infection and related risk factors among young Thai men between 2010 and 2011

PLOS ONE

Julius Eleazar dC. Jose, Boonsub Sakboonyarat, Khunakorn Kana, Thippawan Chuenchitra, Akachai Sunantarod, Supanee Meesiri, Mathirut Mungthin, Kenrad E. Nelson, Ram Rangsin

Reviewer #1: 

In Methods :

1. Please add inclusion and exclusion criteria of the subject

►Response: “Inclusion criteria consisted of men who (a) were 18 years of age or older and (b) gave informed consent. Exclusion criteria comprised the new RTA conscripts who were unable to answer the self-administrated questionnaire during the specified dates in each military camp” [Page 7, Line 123-126].

2. Please add operational definition of subject characteristics especially definition about the MSM and history of : IVDU Incarceration, previous HIV Testing, Blood transfusion, sex with a female sex worker, number of lifetime sex partner, sex with another man, sexual experience, sexual preference, sexual transmitted infection, sex in exchange for gifts, sexual coercion, whether in 3, 6 or 12 months.

►Response: We add the sentences, “ Men having sex with men status was defined as the lifetime sexual activity of a man having sex with another man and not as the identity expressed by the person. Lifetime sex partner consisted of the number of sex partners including the types of sexual partners. History of intravenous drug use, history of incarceration, history of previous HIV testing, history of sex with a FSW, history of sex in exchange for gifts/money, history of sexual coercion, and sexual preference were all defined as the occurrence of the specific events in their lifetime. Sexual experience including the classification of MSM status were inferred from the responses to different but related item in the questionnaire. Sexual transmitted infection was defined as the participants having a history of STIs in the previous 12 months before induction. Condom use with a female sex worker was defined as the participants having a history of condom use with a FSW in the last 12 months” in the Methods section[Page 8, Line 139-150].

In Discussion:

1. Maybe you need more explain why having sex with a female sex worker have protective for the HIV risk (table. 2)

►Response: We have rewritten the sentence in the results section explaining the table 2 results from “Having a history of sex with an FSW (AOR: 0.58; 95% CI: 0.43 – 0.77) was found to be a protective risk factor for HIV infection in the overall study population.” to “Having a history of sex with an FSW (AOR: 0.6; 95% CI: 0.4 – 0.8) was inversely associated with HIV infection in the overall study population.” [Page 12,line 208-210]

We found that having a history of sex with a FSW was inversely associated with HIV infection among overall study participants (AOR: 0.6; 95%CI 0.4 – 0.8). Additionally, when we analyzed the data to identify the associations between having a history of sex with a FSW and HIV infection stratified by MSM status, we found that in the non-MSM study population, having a history of sex with a FSW was no longer associated with HIV infection in 2010 and 2011 (AOR: 1.1; 95% CI: 0.8 – 1.6), while having a history of sex with a FSW was still inversely associated with HIV infection in an MSM subgroup (AOR: 0.3; 95%CI 0.2 – 0.5). One of the explanations for this finding is that having a history of sex with an FSW may be a proxy indicator for having a heterosexual preference in our study population. This explanation is also supported by our finding in Table 4 (Risk factors to HIV infection among identified MSM). The prevalence of HIV infection among those MSM who reporting expressing sexual desire with females only was 0.7%, while it totaled 19.1% among those MSM reporting sexual desire with males only. 

Characteristics Total HIV+ (%) Crude Odds Ratio (95% CI)

Sexual preference/desire 

 Female only 3,735 27 (0.7) 1.0

 Male only 288 55 (19.1) 32.4 (20.1 – 52.4)

 Both male and female 504 39 (1.7) 11.5 (7.0 – 19.0)

We add the sentence, “We found that having a history of sex with a FSW was inversely association with HIV infection among overall study participants. When we analyzed the data to identify the association between a history of sex with a FSW and HIV infection stratified by MSM status, we found that in the non-MSM study population, having a history of sex with a FSW was no longer associated with HIV infection in 2010 and 2011, while a history of sex with a FSW was still inversely associated with HIV infection in the MSM subgroup. 

One of the explanations for this finding is that having a history of sex with an FSW may be a proxy indicator for having a heterosexual preference in our study population. This explanation is also supported by our findings that the prevalence of HIV infection among those MSM who reporting that they had sexual desire with females only was 0.7%, while it totaled 19.1% among those MSM reporting sexual desire with males only” in the Discussion section.[Page 21, line 292-303] 

2. Why you concern about the non-inclusion of transgender women as the limitation of this study and it need to stated clearly in discussion.

►Response: We have rewritten the sentence “Exemptions are given to a small subset of men who are either disabled or severely ill, transgender women and individuals who participate in alternative military service including the Thai Reserve Officer Training Corps Student (TROTCS) program.” in the Methods section. [Page 5, Line 86-89 ] 

We have removed the sentence, “One of the limitations of the present study was the non-inclusion of transgender women because they were not selected during the lottery process” from the Discussion section. 

Reviewer #2: 

Abstract

1. Second sentence, “The proportion of….” seems misplaced. I know you want to highlight this data, but it seems premature to place it second, whereas you repeat the same point below in the 2nd half of results.

►Response: We have removed the sentence “The proportion of HIV positive men, who reported having had sex with another man, was 40.20%.”from the abstract. [Page 2]

2. What is the level of significance between urban and rural? Important for the abstract.

►Response: We have rewritten the sentence “The prevalence of HIV infection among MSM in urban (2.8%) and rural (2.4%) areas were relatively comparable (p-value = 0.44).” in the abstract.

3. First sentence of Conclusion does not accurately capture the overall point of the study.

►Response: We have rewritten the conclusion section “The prevalence of HIV infection among young Thai men has continued to remain below 0.5% during 2010 and 2011. High risk sexual activity, including MSM, played a major role in the HIV epidemic in this population. Effective HIV prevention programs should cover MSM expressing heterosexual desire as well as having sex in exchange for gifts/money and be implemented in both urban and rural areas.” was added to the abstract section [Page 3, Line 30-34]

Background

1. Human immunodeficiency virus should NOT be capitalized

►Response: We have rewritten the word “human immunodeficiency virus” in the background section. [Page 4, Line 41 ]

2. Line 53: in whom is that data reported on?

►Response: We have rewritten the sentence, “Between 1991 and 1995, the proportion of young Thai military conscripts in northern Thailand having sex with an FSW dropped from 81.4% to 63.8%; condom use increased from 61% to 92.6%, while HIV prevalence decreased to 6.7% from a high of 12.5% [6-9]” in the Introduction section [Page 4, Line 53-56].

3. Line 65: very nonspecific, what does it mean to be “a shift to transmission”? This data is available and can be expressed as a proportion.

►Response: We have removed the sentence, “However, a shift to

transmission among men having sex with men (MSM) has been observed” from the introduction section. 

We added the sentence, “However, HIV transmission among MSM is growing rapidly. In 2005, this transmission route contributed 22.6% of new infections and would be 33% in 2010 and 43% by 2015.” in the Introduction section [Page 5, Line 66-68].

4. STI is not capitalized either

►Response: We have rewritten the word “sexual transmitted infection” in the background section [Page 5, Line 74].

Methods

1. First sentence is passive, needs to be more active tense and specific

►Response: We have re-written the sentence, “The RTA holds an annual selection of young men aged 21 years for conscription in April at the district level of their home province.” in the Methods section. [Page 5, Line 85-86]

2. How can conscripts answer any of the questions described?

►Response: We added the sentences, “After the written informed consent process, the enrolled study participants were asked to complete self-administered questionnaires in the private environment in their camps during the first 8 weeks of the basic military training. These questionnaires were completed before the HIV test results were reported to the men to avoid information bias. All completed questionnaires were sent directly from each training unit to the data management unit in Bangkok. The questionnaire was used to identify risk factors to HIV infection that were of interest and has been a developed from related risk factors studies among young Thai men” in the Methods section. [Page 7-8, Line 130-139]

3. “Living alone or with friends” are quite different. What is the other option? Family?

►Response: We have rewritten the living status in Tables 1, 2, 3 and 4. 

Table 1

Living with familya n (%)

 No 2,228 (3,6)

 Yes 60,293 (96.4)

Table2

Characteristics Total HIV+ (%) Crude OR

(95% CI) Adjusted OR

(95% CI)

Living with familya 

 No 2,228 28 (1.3) 2.8 (1.9 – 4.2) 1.5 (1.0 – 2. 5)

 Yes 60,293 267 (0.4) 1.0 1.0

Table 3

Characteristics MSM (%)

N=4,589 Non-MSM (%)

N=53,845 p-value

Living with familya <0.001c

 No 249 (5.6) 1,805 (3.4) 

 Yes 4,236 (94.4) 51,153 (96.6) 

Table 4

Characteristics Total HIV+ (%) Crude OR

(95% CI) Adjusted OR

(95% CI)

Living with familya 

 No 249 15 (6.0) 2.5 (1.4 – 4.4) -

 Yes 4,236 105 (2.5) 1.0 -

aLiving with family; parents, wife/lover and relatives. 

Results:

1. Need data about the number of times each male had ever been HIV tested.

►Response: Unfortunately, we did not include the number of times each male had ever been HIV tested. The questionnaires provided the information only about history of previous HIV testing and test results (lifetime and the last 12 months). 

We added the sentence “History of previous HIV testing in lifetime and previous 12 months accounted for 26.8% and 8.2%, respectively” in the 

Results section [Page 11, Line 193-194].

2. Data needs to be drilled down, in other words those men who have a same sex experience, how often was that? When was the last time they had MSM experience? How frequent was the contact?

►Response: We did not have the information related to frequency of last time sexual contact with the sexual partners of the study participants. However, we have the information on the number of sexual partners in the last 12 months. 

We added the sentence, “Of the 4,594 MSM, 31.8% of them reported a same sex experience in the previous 12 months. The median number of male sex partners was 2 individuals, while the median number of male sex partners in their lifetime was 3 individuals” in the results section [Page 17, Line 227-229].

3. Re-order the results to be from strongest association to weakest.

 ►Response: We have rewritten the sentences, “Risk factors for HIV infection are summarized in Table 2. The risk factors that were independently associated with HIV infection among the young Thai men includes having exclusively homosexual (AOR: 14.4; 95%CI: 10.4 – 19.8) or bisexual activity (AOR: 3.7; 95%CI: 2.4 – 5. 6), having no formal education (AOR: 6.5; 95%CI: 2.3 – 18.4) or a bachelor’s degree (AOR: 1.8; 95%CI: 1.0 – 3.0), having a history of STI (AOR: 2.3; 95%CI: 1.6 – 3.3), providing sex in exchange for gifts/money (AOR: 2.0; 95%CI: 1.5 – 2.8), and being single (AOR: 1.6; 95% CI: 1.1 – 2.2) after adjusting for age, living status and region of residence 2 years before induction. Having a history of sex with an FSW (AOR: 0.6; 95% CI: 0.4 – 0.8) was found to have an inverse association to HIV infection in the overall study population” [Page 12, Line 201-210].

“Risk factor analysis for HIV infection among MSM is summarized in Table 4. The identified risk factors for HIV infection among MSM includes history of versatile (AOR: 4.7; 95% CI: 2.9 – 7.5) or exclusively receptive (AOR: 3.6; 95% CI: 1.6 – 7.7) anal sex (as compared with exclusively insertive), living in the western region 2 years before induction (AOR: 3.5; 95% CI: 1.2 – 10.4), higher educational attainment (AOR: 2.7; 95% CI: 1.2 – 5.7), and history of sex in exchange for gifts/money (AOR: 2.3; 95% CI: 1.5 – 3.5). Having a history of sex with an FSW (AOR: 0.3; 95% CI: 0.2 – 0.5) was found to have an inverse association to HIV infection.” in the Results section. [Page 17, Line 231-238]

4. Discuss HIV infection prevalence in a separate paragraph.

►Response: We have rewritten HIV infection prevalence in a separate paragraph.

“Prevalence of HIV infection 

 A total of 301 (0.5%) young Thai men were identified to be HIV infected. Of 4,594 identified MSM, 121 (2.6%) were identified to be HIV infected cases. This corresponded to 40.2% of the total number of HIV cases in the study. ” [Page 12, Line 196-199]

5. Line 173 and following: where is the 7.86% data about a history of sex with another man?

►Response: We found that 4,589 men reported a history of sex with another man (7.9% of total participants)[Table 1]. 

The 4,589 MSM can be categorized by lifetime sexual experience in 2 groups including bisexual (4.8% of all participants) and exclusively homosexual (3.1% of all participants). The data is shown in Table 1.

Table 1. Demographic and behavioral profile of the participants before induction.

Characteristics N = 63,667 (%)

History of sex with another man 4,589 (7.9)

Sexual experience (Lifetime) 

 Exclusively heterosexual 53,845 (92.2)

 Bisexual 2,781 (4.8)

 Exclusively homosexual 1,808 (3.1)

The data related to 4589 MSM (7.9% of all participants) is shown again in table 2. We have rechecked the number presented in table 2 and rewritten as follows. 

Table 2. Univariate and multivariate analysis of risk factors to HIV infection among the participants inducted in the RTA in November 2010 and May 2011

Characteristics Total HIV+ (%) Crude Odds 

Ratio (95% CI) Adjusted Odds

Ratio (95% CI)

Sexual experience 

Exclusively heterosexual 53,845 167 (0.3) 1.0 1.0

Bisexual 2,781 33 (1.2) 3.9 (2.7 – 5.6) 3.7 (2.4 – 5.6)

Exclusively homosexual 1,808 88 (4.9) 16.4 (12.7 – 21.4) 14.4 (10.4 – 19.8)

6. I would not call sex with an FSW “protective” but rather has an inverse association.

►Response: We have rewritten from “with an FSW (AOR: 0.58; 95% CI: 0.43 – 0.77) was found to be a protective risk factor for HIV infection” to “Having a history of sex with an FSW (AOR: 0.3; 95% CI: 0.2 – 0.5) was found to have an inverse association to HIV infection” in the results section. 

7. Paragraph starting at line 200: too much detail, regurgitation of the tables.

►Response: We have removed the paragraph which has too much detail and regurgitation of the tables.

Discussion

1. Valid discussion points

2. The association between education and risk through social media is tenuous at best, with no evidence. “Might be enhanced” is not evidence. Please see the findings from this study among Thai MSM: Piyaraj P, van Griensven F, Holtz TH, Nelson K. The impact of internet use to recruit sex partners, and methamphetamine use on incident HIV infection among men who have sex with men, Bangkok, Thailand. Lancet HIV 2018;5(7):e379-e389. https://doi.org/10.1016/S2352-3018(18)30065-1. PMCID 6452023.

►Response: We agree with the reviewer. 

An alternative explanation of the results may be discussed below. 

Because the majority (82.4%) of the MSM in our study population expressed having sexual preference/desire with females only. In our study, sexual preference was a relatively strong risk factor for HIV infection (with male only crude OR 52.3; 95%CI 38.1 – 71.7, with both male and female crude OR 22.3; 95%CI 15.9 – 31.2).

We analyzed data to obtain associations between college degree graduated and sexual preference (overall participants, MSM, and non-MSM); the results are shown in Tables A, B and C.

The higher risk for HIV infection among those graduating with college might be because those participants had a higher proportion (4.1% among total participants, 58.6% in the MSM subgroup) of sexual preference with males (male only or both males and females) compared with those graduating with less than a college degree (1.5% among total participants, 15.6% in the MSM subgroup).

This effect was also shown in the MSM subgroup population (with male only crude OR 32.4; 95%CI 20.1 – 52.4, with both male and female crude OR 11.5; 95%CI 7.0 – 19.0). 

Table A. Association between sexual preference and graduating with a college degree graduated among all participants

 College degree graduated

Sexual Preference No n(%) Yes n(%)

Female 58203 (98.5) 3508 (95.8)

Male 316 (0.5) 78 (2.1)

Both 586 (1) 75 (2)

Table B. Association between sexual preference and college degree graduated among MSM

 MSM

 College degree graduated

Sexual Preference No n(%) Yes n(%)

Female 3639 (84.4) 87 (41.4)

Male 224 (5.2) 68 (32.4)

Both 449 (10.4) 55 (26.2)

Table C. Association between sexual preference and college degree graduated among non-MSM

NON-MSM

 College degree graduated

Sexual Preference No n(%) No n(%)

Female 50019 (99.7) 3082 (99.6)

Male 55 (0.1) 3 (0.1)

Both 90 (0.2) 10 (0.3)

We have rewritten the discussion section, “We identified that educational attainment was independently associated with HIV infection both in the total study population and the MSM subgroup. Those men who had obtained a bachelor’s degree or higher were more likely to acquire HIV infection than those who were at grade 9 and lower educational level in both the total study population of young Thai men and the MSM subgroup. Our finding was consistent with the related study from 2005-2009 reporting that the young Thai men with college degrees had a higher risk for HIV infection. [Rangsin, 2015] The higher risk for HIV infection among college graduated might be because the participants with college degrees had a higher proportion of sexual preference with males (4.1% in total participants, 58.6% in the MSM subgroup) compared with those with less than college degree (1.5% among total participants, 15.6% in the MSM subgroup). In our study, sexual preference was a relatively strong risk factor for HIV infection (with male only crude OR 52.5; 95%CI 38.1 – 71.7, with both male and female crude OR 22.3; 95%CI 15.9 – 31.2). This effect was also shown in the MSM subgroup population (with male only crude OR 32.4; 95%CI 20.1 – 52.4, with both male and female crude OR 11.5; 95%CI 7.0 – 19.0).” [Page 21, Line 304-319] 

3. Were the participants randomly selected? Or were they part of a population-based sample of young men in Thailand? Any possible source of bias? 

►Response: Our study population comprised all new conscripts inducted in the military service of the Royal Thai Army in November 2010 and May 2011. The RTA holds an annual selection of young men aged 21 years for conscription in April at the district level of their home province. Those chosen will enter military service in either May or November of the same year. 

The conscripts consist of 2 groups including (1) conscripts by lottery system and (2) military volunteers, accounting for 69.3% and 30.7%, respectively, in our study period. The first group was randomly selected from the eligible young Thai men aged 21 years old at district level nationwide. Exemptions are given to a subset of men who are either disabled or severely ill, transgender women (TG) and individuals who participate in alternative military service. The Royal Thai Army also allows young Thai men aged 21 years old to participate in the army as military volunteers, the second group. 

One of the reasons for exemption from the conscription process by lottery system is alternative military service as a Thai Reserve Officer Training Corps Student (TROTCS). In Thailand, the government allows approximately 100,000 secondary school and university students to participate in the TROTCS program as an alternative military service each year. Those students who complete the TROTCS program would be excluded from the conscription process when they reach 21 years of age. Because the TROTCS program was provided for only men who are in education institutions; therefore, the new conscripts tended to have a relatively lower educational level than those who were excluded as TROTCS. 

In 2010 and 2011, the total number of young Thai men aged 21 years old was 459,067 and 478,319, respectively. In the November 2010 round of induction, the total number of the new conscripts was 30,694 men and 27,662 (90.1%) men were enrolled in our study. In the May 2011 round of induction, the total number of the new conscripts was 38,123 men and 35,995 (94.4%) men were enrolled in our study. 

To evaluate the potential bias between conscripts by lottery system and by military volunteers, we found that the HIV status among both groups did not differ. 

Table A Association between HIV status and types of induction

 Volunteers Lottery System p-value

 n (%) n (%) 

HIV Status 0.73

Negative 19411 (99.5) 43934 (99.5) 

Positive 95 (0.5) 206 (0.5) 

In terms of the prevalence of HIV infection among young Thai men aged 21 years old, the possibility of a bias related to the prevalence estimate was relatively low as shown in Table A.

Table B. Association between HIV status and types of induction among MSM

 Volunteers Lottery System p-value

 n (%) n (%) 

HIV Status 0.002

Negative 1497 (98.4) 2971 (96.8) 

Positive 24 (1.6) 97 (3.2) 

However, the prevalence of HIV infection among young Thai men aged 21 years old reporting a history of sex with another man (MSM) may have been underestimated. In the study population group from the lottery system, the prevalence of HIV infection was 3.2% while it totaled 1.6% among military volunteers. In addition, those MSM with higher education and young men who were excluded from the conscription process due to an alternative military service as TROTCS may have a higher prevalence of HIV infection. 

Even though, we did not have information on this group (MSM in TROTCS), we think that this group may had a higher prevalence of HIV infection due to the higher education level. Those MSM with higher educational level may have a lower proportion of heterosexuals reporting sex with another man (usually due to sex in exchange for gift/money) compared with those MSM with lower educational level. 

 

We added the sentence “one of the limitations in our study related to the possibility of an underestimated prevalence of HIV infection among MSM. In Thailand, the military conscription process excluded the young Thai men who participating in an alternative military service as the Thai Reserve Officer Training Corps Student (TROTCS), which involves approximately 20% of the total cohort of 21-year-old Thai men each year. Even though, we did not have information on this MSM group in TROTCS. However, this group may have a higher prevalence of HIV infection because they had a higher education level. Those MSM with higher educational level may have had a lower proportion of heterosexuals reporting having sex with another man compared with those MSM with lower educational level. The heterosexual men who had experienced sex with another man had a lower risk for HIV infection. Some of the heterosexual men may have had sex with another man in exchange for gift/money .” in the Discussion section. [Page 23, line 332-343]

4. Also, they were not selected young men and MSM<rather just “men”?

►Response: We have rewritten the sentence “the prevalence and risk factors for HIV infection in a randomly selected national sample of young men conscripted in the RTA in November 2010 and May 2011” in the Conclusion section.[Page 23-24, line 352-354] 

- Minor Essential Revisions

1. In Discussion, “reported” or “report” in this manuscript? Since you are reporting now, would suggest not using the past tense. This holds true throughout the Discussion, i.e. line 271, etc

►Response: The term reported was converted to report as suggest. 

2. Discussion: sex between men LIKELY played a role….

►Response: We have rewritten the sentence “From our current study, we found that sex between men likely played a major role in the recent HIV epidemic among young Thai men.” [Page 20, Line 255-256]

3. Prefer “HIV-infected” instead of “HIV-positive”

►Response: We have rewritten from “HIV-positive” to “HIV-infected”

4. Line 274: Diverted sexual partners, unclear meaning

►Response: We have rewritten the sentence, “Notably, while the proportion of young Thai men who reported having sex with an FSW had declined to less than 40% in 2009, the proportion of men who reported having sex with a girlfriend increased”. [Page 21, Line 288-290] 

5. Spelling of Acknowledgments

►Response: We have rewritten from “Acknowledgements” to “Acknowledgments”

- Discretionary Revisions

These are suggested improvements, which the author may choose to consider.

1. Although you have the degrees of freedom to report 2 digits to the right of the decimal, this level of detail does not really aid the reader, 40.2% is as useful as 40.19%.

►Response: The decimal places were decreased to one decimal place except for the p-value. 

Sincerely Yours,

Ram Rangsin, M.D., M.P.H., Dr.P.H.

---

## [Decision Letter · Decision Letter 1]

28 Jul 2020

PONE-D-20-03591R1

Prevalence of HIV infection and related risk factors among young Thai men between 2010 and 2011

PLOS ONE

Dear Dr. Rangsin,

Thank you for submitting your manuscript to PLOS ONE. After careful consideration, we feel that it has merit but does not fully meet PLOS ONE’s publication criteria as it currently stands. Therefore, we invite you to submit a revised version of the manuscript that addresses the points raised during the review process.

Thank you for this timely revision and attending to the reviewer's comments. There a just a few very minor issues that need to be addressed per one of the reviewers below. 

We look forward to receiving your revised manuscript.

Kind regards,

Ethan Morgan

Academic Editor

PLOS ONE

Reviewers' comments:

Reviewer's Responses to Questions

**Comments to the Author**

1. If the authors have adequately addressed your comments raised in a previous round of review and you feel that this manuscript is now acceptable for publication, you may indicate that here to bypass the “Comments to the Author” section, enter your conflict of interest statement in the “Confidential to Editor” section, and submit your "Accept" recommendation.

Reviewer #1: All comments have been addressed

Reviewer #2: (No Response)

2. Is the manuscript technically sound, and do the data support the conclusions?

Reviewer #1: Yes

Reviewer #2: Yes

3. Has the statistical analysis been performed appropriately and rigorously? 

Reviewer #1: Yes

Reviewer #2: Yes

4. Have the authors made all data underlying the findings in their manuscript fully available?

Reviewer #1: Yes

Reviewer #2: Yes

5. Is the manuscript presented in an intelligible fashion and written in standard English?

Reviewer #1: Yes

Reviewer #2: Yes

6. Review Comments to the Author

Reviewer #1: I have read the all the responses for this manuscript and I agree with all corrections. No need more corrections from me. I suggest to accept this manuscript for publication. Thank you.

Reviewer #2: Thank you for your response - and changes. Explanations are clear. Data looks solid, epi claims are now revised.

Have not changed to "HIV-infected" throughout, just in select places.

Also, "human immunodeficiency virus" is still in caps in some places. should not be.

7. PLOS authors have the option to publish the peer review history of their article (what does this mean?). If published, this will include your full peer review and any attached files.

Reviewer #1: **Yes: **I Ketut Agus Somia

Reviewer #2: **Yes: **Timothy H Holtz

---

## [Author Response · Author response to Decision Letter 1]

29 Jul 2020

PONE-D-20-03591R1

Prevalence of HIV infection and related risk factors among young Thai men between 2010 and 2011

PLOS ONE

Julius Eleazar dC. Jose, Boonsub Sakboonyarat, Khunakorn Kana, Thippawan Chuenchitra, Akachai Sunantarod, Supanee Meesiri, Mathirut Mungthin, Kenrad E. Nelson, Ram Rangsin

Reviewer #2: 

1. Thank you for your response - and changes. Explanations are clear. Data looks solid, epi claims are now revised. Have not changed to "HIV-infected" throughout, just in select places.

Also, "human immunodeficiency virus" is still in caps in some places. should not be.

►Response: “We have rewritten the words “human immunodeficiency virus”[Page 3, Line 2]and “HIV-infected ”[Page 4, Line 63 & Page12, Line 197-198] 

Sincerely Yours,

Ram Rangsin, M.D., M.P.H., Dr.P.H.

---

## [Editor Report · Decision Letter 2]

31 Jul 2020

Prevalence of HIV infection and related risk factors among young Thai men between 2010 and 2011

PONE-D-20-03591R2

Dear Dr. Rangsin,

We’re pleased to inform you that your manuscript has been judged scientifically suitable for publication and will be formally accepted for publication once it meets all outstanding technical requirements.

Kind regards,

Ethan Morgan

Academic Editor

PLOS ONE
---

## [Editor Report · Acceptance letter]

6 Aug 2020

PONE-D-20-03591R2 

Prevalence of HIV infection and related risk factors among young Thai men between 2010 and 2011 

Dear Dr. Rangsin:

I'm pleased to inform you that your manuscript has been deemed suitable for publication in PLOS ONE. Congratulations! Your manuscript is now with our production department. 

Kind regards, 

on behalf of

Dr. Ethan Morgan 

Academic Editor

PLOS ONE